# Autonomous Heading Planning and Control Method of Unmanned Underwater Vehicles for Tunnel Detection

**Tianxing Xia [1] , Dehao Cui [2], Zhenzhong Chu [3,\*] and Xing Yu [1]**

[1] Logistics Engineering College, Shanghai Maritime University, Shanghai 201306, China; 18971276707@163.com (T.X.); 202130210009@stu.shmtu.edu.cn (X.Y.)
[2] Shenzhen Dongjiang Water Source Project Management Office, Shenzhen 518172, China; 15013531192@163.com
[3] School of Mechanical Engineering, University of Shanghai for Science and Technology, Shanghai 200093, China
[\*] Correspondence: zhenzhongchu@usst.edu.cn; Tel.: +86-021-5527-0456

**Abstract:** To address the challenge of unmanned underwater vehicle (UUV) autonomous navigation in long-distance underwater tunnel detection tasks and improve the control performance of its heading control system, a method of autonomous heading planning and control based on sonar-ranging feedback control was proposed. This method combines UUV's autonomous heading planning technology with the heading proportion-integral-derivative (PID) control algorithm, optimizing the acquisition method of controller input data, to impart specific adaptive characteristics to the controller. Using the ranging principle of ultrasonic spontaneous self-collection, it is possible to obtain the yaw direction and angle of the vehicle relative to the target heading in the tunnel and continuously adjust the control law to change the heading as the vehicle's heading status changes during navigation. The effectiveness of the autonomous heading planning and control method is verified through pool experiments. The analysis and experimental results show that the proposed heading planning method achieves good control effect in UUV's underwater tunnel detection heading control, and exhibits obvious advantages in long-distance closed tunnel environments. UUV can adaptively adjust the heading according to the tunnel environment and has a fast response and strong applicability in planning and controlling the heading.

**Keywords:** UUV; heading control; autonomous heading planning; ranging sonar; ROS

## 1. Introduction

Water conveyance tunnels are a common feature in water conservancy projects, and their quality directly impacts the entire project and plays a vital role in promoting national economic development and ensuring public welfare [1]. Therefore, regular safety inspections and post-disaster repairs for these tunnels are essential. However, the long tunnel length, severe tunnel environment, and continuous water flow make manual inspection hazardous. Thus, there is an urgent need for UUV technology dedicated to tunnel detection. In complex tunnel environments with high flow velocities, designing a corresponding heading planning and control method to achieve stable vehicle navigation control is a critical problem [2].

The autonomous navigation problem of underwater vehicles in tunnels mainly includes two parts: heading guidance and control. Heading control is the most basic control execution module for achieving autonomous navigation of UUV, which can directly determine the overall control effect of planning control strategies. Standard heading control methods include heading PID control, fuzzy control, adaptive control, sliding mode control, and neural network methods [3]. Solve the problem of heading control in ocean vehicles, and many researchers have attempted to solve this problem. Wan et al. used the PID control method to design a heading controller for teleoperated underwater vehicles and

proved its feasibility through experiments [4]; Liu et al. proposed a fuzzy control method for heading keeping based on least squares support vector machines [5]; a robust adaptive heading keeping control method based on Lyapunov stability theory by Zhu et al. [6]; Vu MT et al. designed a state guarantee method for underwater vehicles based on a robust station guaranteed (SK) control algorithm based on sliding mode control (SMC) theory [7]; and Luo et al. designed a neural network heading keeping control method based on the backstepping method [8].

The most important indicator of heading control effectiveness is the stability and robustness of its control. Sliding mode control technology has strong robustness because the characteristics and parameters of the system only depend on the designed switching hyperplane and are not related to external interference [9]. Therefore, it is also widely used in the stability research of marine vehicles or their second-order systems. Currently, the most commonly used methods include dynamic sliding mode control, robust sliding mode control, multiple sliding mode control [10], nonsingular sliding mode control [11,12], and terminal sliding mode control [13]. In addition, many researchers have also made attempts to study this aspect. Kurniawan proposed an improved design strategy for repetitive sliding mode controllers, which can be used for heading control to accurately track reference signals [14]; Mashhad designed $H_\infty$ using the linearized autonomous underwater vehicle (AUV) model Robust controller and simulation to verify its effectiveness for AUV heading control [15]; and Zhang Q. et al. proposed a nonlinear fuzzy control algorithm for ship heading keeping based on feedback linearization using the approximation ability of the fuzzy system constructed by radial basis function neural networks [16]. The above research is only limited to the study of heading control algorithms. On the one hand, these methods often rely on the model of the system, which is not sufficiently inclusive of model uncertainty. On the other hand, they usually involve complex computational processes. Introducing algorithms with high computational complexity can lead to explosive increases in the computational complexity of heading control algorithms and even conflicts between different planning and control modules [17].

In addition, these heading control algorithms proposed above do not effectively solve the problem of UUV heading guidance in tunnels. To solve such autonomous navigation operations, it is usually necessary to plan an optimal path from the starting point to the destination, and then use path tracking and other methods to complete the autonomous navigation of UUV [18]. Mai et al. proposed a method for generating AUV course trajectory based on line-of-sight algorithm (LOS), and designed a dynamic positioning control system based on motion control and allocation control [19]. However, the LOS algorithm for heading planning depends on the position of the target point on the reference path, making it difficult to accurately measure the coordinate position of the UUV pair in practical engineering applications, making it challenging to ensure the control effect; Wang et al. combined nonlinear model prediction with neural networks to study the trajectory tracking problem of underwater vehicles [20]. Model predictive control ensures control effectiveness by predicting future states. However, due to excessive dependence on system models, it is difficult to implement systems with uncertain models and complex calculations; Zhang H. et al. designed a new dynamic path re-planning algorithm, AT&x002A, to solve the motion path planning problem [21]. However, it is easy to encounter the problem of target loss in obstacle avoidance strategies and path planning solutions. Therefore, designing a heading planning and control method with low computational complexity, simple structure, and excellent control effect and applicability is critical to achieving autonomous heading planning and control for UUV.

This paper studies the autonomous heading planning and control method used in tunnel detection for UUV. Based on the above issues, an autonomous heading planning method based on sonar-ranging feedback control is proposed. Combining it with the heading PID control method, a method for autonomous heading planning and control is designed. This method uses a simple structured heading control method to reduce computational complexity. At the same time, the algorithm for heading guidance weakens

the dependence on the system model and external parameters, ensuring applicability and control effectiveness. Finally, a pool experiment Experimental verification is conducted on the UUV platform designed based on the robot operating system (ROS). The contributions made in this paper are summarized as follows:

(1) The open-source underwater vehicle platform BlueROV is modified, and the control system is rebuilt. The ranging sonar and compass are added as the heading feedback sensor, which is used as the experimental platform for the heading planning method proposed;

(2) Based on the principle of ultrasonic auto-receive ranging, a new heading planning method based on ranging sonar feedback control is proposed, and UUV's autonomous heading planning control is realized.

The article is structured as follows: Section 2 introduces the problems studied in this paper, including the hardware framework and control system based on ROS design of the heading planning of the UUV. Section 3 introduces the theory of autonomous heading planning and control methods. Section 4 is the relevant test experiment, including the introduction of the test environment and the experimental scheme, and the analysis of experimental results. The last section comprehensively describes the relevant conclusions of the experiment and the further work to be carried out in the future.

## 2. Problem Description

### 2.1. Mathematical Model

#### 2.1.1. Assumptions

This paper studies the control method for autonomous heading planning of the UUV for tunnel detection. The vehicle can independently plan the heading deflection angle to the target point based on its position in the tunnel and achieve the yaw motion through a heading PID controller. The entire process mainly involves controlling the movement of UUV in the horizontal plane. To clarify the research issues, we now make the following assumptions and constraints:

- Mainly studying the heading control of UUV in the horizontal plane, ignoring the roll, trim, and snorkeling motions of UUV, and simplifying the model to a degree of freedom model;
- Due to the low advance velocity of UUV during tunnel detection, the impact of water flow speed on sonar ranging is not considered, and the effect of water quality in the simulation and test environment on sonar ranging is not considered;
- To better represent the heading in the algorithm design, this paper abstracts UUV into a straight line for mathematical modeling, ignoring the deviation angle between the theoretical model heading and the actual physical model heading;
- Do not consider the impact of gravity and buoyancy on UV motion in the vertical plane, and ignore the restoring force generated during their motion;
- The measurement range of a sonar ranging sensor is limited, and the maximum range constraint is considered in data feedback;
- The thrust that UUV thrusters can generate is limited, and the thrust distribution on the horizontal plane takes into account the constraint of thrust saturation.

We will establish and derive the following mathematical model based on the above assumptions and constraints.

#### 2.1.2. Kinematic and Dynamic Model

The control objectives of the UUV during its movement process include its position, posture, speed, angular velocity, etc., involving six degrees of freedom of motion control—namely, the surge, sway, heave, yaw, pitch, and roll of the UUV. To better describe these motion characteristics, in engineering applications, they are often artificially decoupled into two types: horizontal plane control and vertical plane control [22]. Establish the UUV coordinate system as shown in Figure 1.

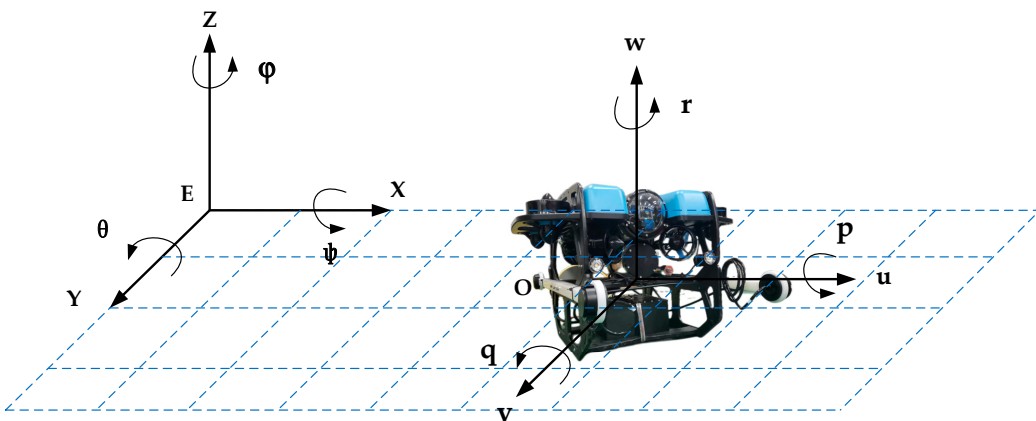

**Figure 1.** Six-degree-of-freedom model of UUV.

In the figure, E represents the geodetic coordinate system, and O represents the carrier coordinate system of the UUV body. The motion of an underwater robot in each degree of freedom corresponds to a variable in both coordinate systems; the corresponding relationship is shown in Table 1.

**Table 1.** Correspondence table of the two coordinate systems of the kinematic model.

| Degree of Freedom of UUV | Position/Posture (E) [1] | Velocity (O) [2] |
|:---:|:---:|:---:|
| Movement-X [3] | $x$ | $u$ |
| Movement-Y [3] | $y$ | $v$ |
| Movement-Z [3] | $z$ | $r$ |
| Rotation-X [4] | $\varphi$ | $p$ |
| Rotatio-Y [4] | $\theta$ | $q$ |
| Rotation-Z [4] | $\psi$ | $r$ |

[1] Position and posture of UUV in geodetic coordinate system; [2] the velocity of UUV in the carrier coordinate system; [3] movement of UUV in the X, Y, and Z directions; [4] and rotation of UUV around X, Y, Z directions.

The establishment of the kinematics model of UUV mainly relies on the transformation between two coordinate systems, and the transformation relationship between the two can be expressed as Equation (1):

$$\dot{\eta} = J(\eta)\mathrm{v} \tag{1}$$

where $J(\eta)$ is the transformation matrix:

$$J(\eta) = \begin{bmatrix} J_1 & O_{3\times3} \\ O_{3\times3} & J_2 \end{bmatrix} \tag{2}$$

where $J_1$ and $J_2$ are:

$$J_1(\eta) = \begin{bmatrix} \cos\theta\cos\psi & \sin\phi\sin\theta\cos\psi - \cos\phi\sin\psi & \cos\phi\sin\theta\cos\psi + \sin\phi\sin\psi \\ \cos\theta\sin\psi & \sin\phi\sin\theta\sin\psi - \cos\phi\cos\psi & \cos\phi\sin\theta\cos\psi + \sin\phi\cos\psi \\ -\sin\theta & \sin\phi\cos\theta & \cos\phi\cos\theta \end{bmatrix} \tag{3}$$

$$J_2(\eta) = \begin{bmatrix} 1 & \sin\phi\tan\theta & \cos\phi\tan\theta \\ 1 & \cos\phi & -\sin\phi \\ 0 & \sin\phi\sec\theta & \cos\phi\sec\theta \end{bmatrix} \tag{4}$$

Based on the previously assumed conditions, the posture vector of the UUV becomes $\eta = [x \, y \, \psi]^T$. The velocity vector becomes $v = [u \, v \, r]^T$, and the two-dimensional kinematic model of the underwater robot is obtained, as shown in Equation (5).

$$\dot{\eta} = \begin{bmatrix} \dot{x} \\ \dot{y} \\ \dot{\psi} \end{bmatrix} = J(\eta)v = \begin{bmatrix} \cos \psi & -\sin \psi & 0 \\ \sin \psi & \cos \psi & 0 \\ 0 & 0 & 1 \end{bmatrix} \begin{bmatrix} u \\ v \\ r \end{bmatrix} \tag{5}$$

The establishment of a dynamic model of an underwater vehicle requires force analysis. The Newton-Euler method is one of the standard methods for analyzing rigid body dynamics. Underwater robots are mainly subjected to thruster thrust, self-gravity, buoyancy, etc. Under the combined action of these forces, the underwater robot generates a six-degree of freedom motion. Based on assumptions, the dynamic model of the two-dimensional plane is obtained as follows:

$$M\dot{v} + C(v)v + D(v)v + g(\eta) = \tau \tag{6}$$

where, $\tau = [F_u \, F_v \, F_r]^T$ refers to the longitudinal thrust, lateral thrust, and heading moment provided by the propeller; $v = [u \, v \, r]^T$ is the longitudinal velocity, lateral velocity, and heading angular velocity of the UUV; $M = diag(M_x, M_y, M_\psi)$ is the inertial matrix of the UUV containing additional mass, wherein $M_x = m - X_{\dot{u}}$, $M_y = m - X_{\dot{v}}$, $M_\psi = m - X_{\dot{r}}$; $D(v) = diag(X_u, Y_v, N_r) + diag(D_u|u|, D_v|v|, D_r|r|)$ is the resistance matrix; $C(v)$ is the Coriolis force matrix, and the expression is shown in Equation (7); $g(\eta)$ is the restoring force vector generated by gravity and buoyancy, and it is known from the previous assumption that $g(\eta) = 0$.

$$\mathbf{C}(v) = \begin{bmatrix} 0 & 0 & -M_y v \\ 0 & 0 & M_x u \\ M_y v & -M_x u & 0 \end{bmatrix} \tag{7}$$

Expand Equation (6), and the UUV's dynamics model is shown in Equation (8):

$$\begin{aligned} \dot{u} &= \frac{M_y}{M_x} vr - \frac{X_u}{M_x} u - \frac{D_u}{M_x} u|u| + \frac{F_u}{M_x} \\ \dot{v} &= -\frac{M_x}{M_y} ur - \frac{Y_v}{M_y} v - \frac{D_v}{M_y} v|v| + \frac{F_v}{M_y} \\ \dot{r} &= \frac{M_x - M_y}{M_\psi} uv - \frac{N_r}{M_\psi} r - \frac{D_r}{M_\psi} r|r| + \frac{F_r}{M_\psi} \end{aligned} \tag{8}$$

The meanings represented by the parameters in the above Equation (8) are shown in Table 2.

**Table 2.** Correspondence table of meanings of kinetic model parameters.

| Parameter | Unit Symbol | Description |
|---|---|---|
| $m$ | $kg$ | Mass |
| $X_u$ | $N \cdot s/m$ | Linear resistance in $u$ direction |
| $Y_v$ | $N \cdot s/m$ | Linear resistance in $v$ direction |
| $N_r$ | $N \cdot s/m$ | Linear resistance in $r$ direction |
| $X_{\dot{u}}$ | $kg$ | Additional mass in the $u$ direction |
| $Y_{\dot{v}}$ | $kg$ | Additional mass in the $v$ direction |
| $N_{\dot{r}}$ | $N \cdot m \cdot s^2$ | Additional rotational inertia in $r$ direction |
| $D_u$ | $N \cdot s^2/m^2$ | Secondary resistance in $u$ direction |
| $D_v$ | $N \cdot s^2/m^2$ | Secondary resistance in $v$ direction |
| $D_r$ | $N \cdot s^2/m^2$ | Secondary resistance in $r$ direction |
| $I_Z$ | $N \cdot m \cdot s^2$ | Rotational inertia |

### 2.1.3. Thrust Distribution Model

The motion of the UUV in the horizontal plane is primarily controlled by the distribution of thrust among its four horizontal thrusters. Various thrust distribution modes

can enable the vehicle to move surge, sway, and yaw in the horizontal plane [23]. Figure 2 depicts the thrust distribution of the vehicle's thrusters in the horizontal direction.

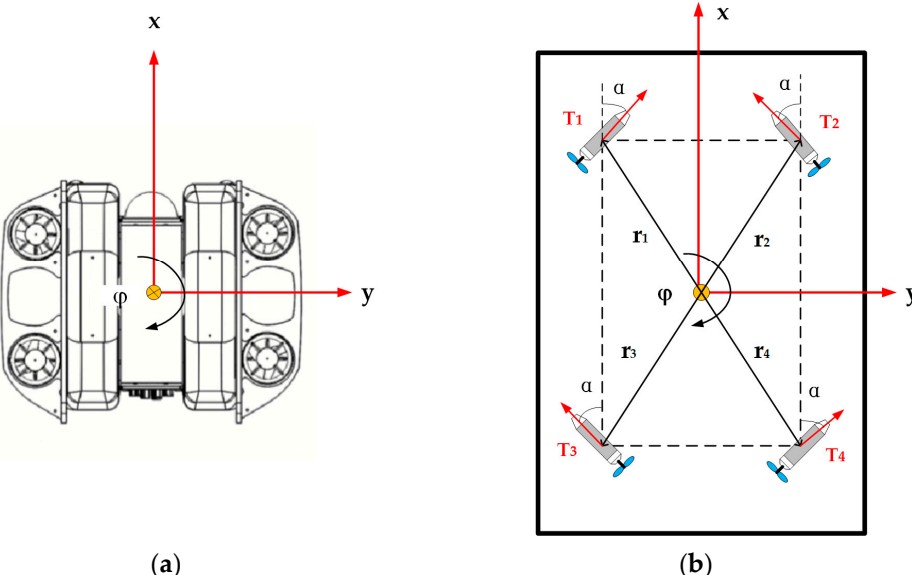

(**a**)                    (**b**)

**Figure 2.** Horizontal thrust distribution diagram of UUV: (**a**) force direction of UUV three degrees of freedom; (**b**) thrust distribution diagram of UUV.

Based on the analysis of the forces acting on the UUV in the horizontal direction, equations can be derived to describe the relationship between the forces/moments and the thrust generated by each thruster.

$$
\begin{aligned}
F_x &= T_1 \cos \alpha + T_2 \cos \alpha + T_3 \cos \alpha + T_4 \cos \alpha \\
F_y &= T_1 \sin \alpha - T_2 \sin \alpha + T_3 \sin \alpha - T_4 \sin \alpha \\
N &= T_1 \times r - T_2 \times r - T_3 \times r + T_4 \times r
\end{aligned}
\tag{9}
$$

In (9), the thrust is expressed by $T_i(i = 1, 2, 3, 4)$, and the included angle between $T_i$ and the X coordinate axis is recorded as $\alpha$. The thrusters are symmetrically distributed, and the distance from each thruster to the vehicle's center of mass is expressed by $r$. The resultant force of the vehicle in the X direction is $F_x$, the resultant force in the Y direction is $F_y$, and the rotational torque in the heading is $N$.

The equation group (9) is expressed in matrix form as:

$$
\begin{bmatrix} F_x \\ F_y \\ N \end{bmatrix} = \underbrace{\begin{bmatrix} \cos \alpha & \cos \alpha & \cos \alpha & \cos \alpha \\ \sin \alpha & -\sin \alpha & \sin \alpha & -\sin \alpha \\ r & -r & -r & r \end{bmatrix}}_{B} \begin{bmatrix} T_1 \\ T_2 \\ T_3 \\ T_4 \end{bmatrix}
\tag{10}
$$

The transformation matrix between the force matrix of the vehicle and the thrust matrix of the thrusters is denoted as matrix $B$. The pseudo-inverse solution of matrix $B$, denoted as $B^{\dagger}$, is used to calculate the relationship between the thrust of the horizontal thrusters and the force in each direction. This allows for the calculation of the thrust of each horizontal thruster corresponding to the rotating torque, enabling control of the UUV in heading planning. The relationship can be expressed as follows:

$$
\begin{bmatrix} T_1 \\ T_2 \\ T_3 \\ T_4 \end{bmatrix} = B^{\dagger} \begin{bmatrix} F_x \\ F_y \\ N \end{bmatrix}
\tag{11}
$$

Given 200 sets of random excitation signals in a certain range in the three degrees of freedom directions in the horizontal plane and using them as the force output in the direction of UUV, a single degree of freedom thrust distribution can be performed on each sampling point to obtain the distribution of the four propellers in the horizontal plane, as shown in Figure 3 below.

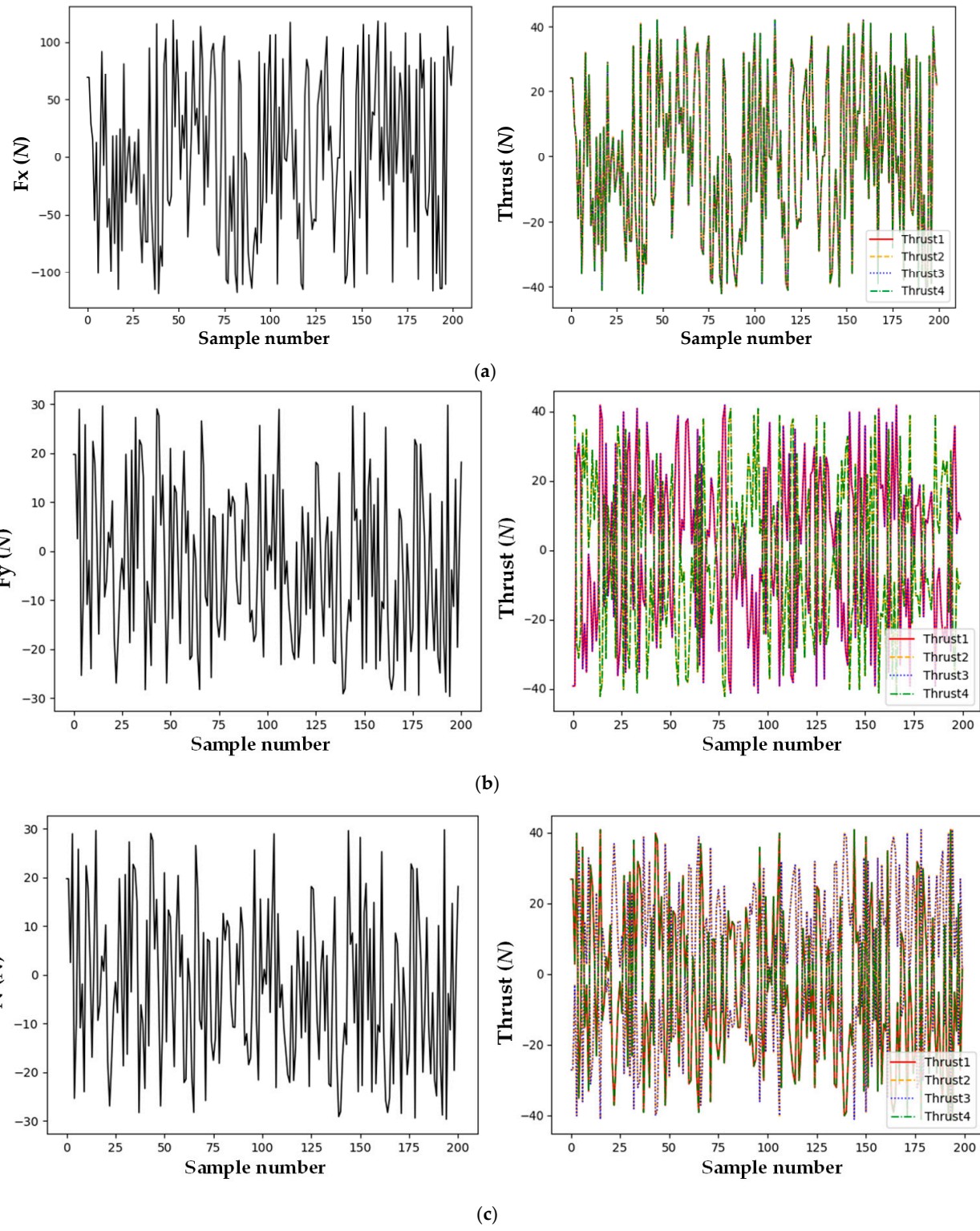

**Figure 3.** Single degree of freedom thrust distribution result: (**a**) thrust distribution result of surge direction; (**b**) thrust distribution result of sway direction; (**c**) thrust distribution result of yaw direction.

According to the constraints assumed in the previous article, there is a range of thrust that each thruster can produce, i.e., $T_i \in [-T_{i\,max}, T_{i\,max}]$. When the UUV performs a combined motion in the horizontal plane, the desired thrust of its three degrees of freedom is outputted simultaneously. This may lead to a few thrusters exceeding the thrust range and the phenomenon of thrust saturation. The result of such thrust distribution is not achieved on the UUV, and to solve such problems, we need to adjust the method of thrust distribution.

In the process of autonomous heading planning, the control of the heading of the UUV is more demanding than the control of the motion in other directions. To solve the thrust saturation problem, we use the heading priority allocation method to ensure the desired control of the UUV's heading by reducing the lateral and longitudinal forces in equal proportion while keeping the bow moment constant. The process is repeated until the thrust of each thruster meets the thrust range. The process is shown in the following Equation (12), where $k$ represents the scaling factor for the reduction.

$$\begin{bmatrix} T_1 \\ T_2 \\ T_3 \\ T_4 \end{bmatrix} = B^\dagger \begin{bmatrix} k \cdot F_x \\ k \cdot F_y \\ N \end{bmatrix} \tag{12}$$

$$s.t.\ T_i \in\ [-T_{i\,max}, T_{i\,max}]\ (i = 1, 2, 3, 4)$$

The thrust distribution method based on heading priority can give priority to ensuring the control effect of the heading when the propeller reaches thrust saturation in the case of multiple degrees of freedom composite motion so that the propeller's thrust can be effectively distributed. The scaling factor $k$ in this article is set to 0.5, and the thrust range of each propeller is $T_i \in [-45N, 45N]\ (i = 1, 2, 3, 4)$. Applying the above three single degree of freedom random excitation signals to the UUV simultaneously, using the heading priority thrust distribution to obtain the thrust of each propeller, it can be seen that the thrust of each propeller is effectively constrained, as shown in Figure 4.

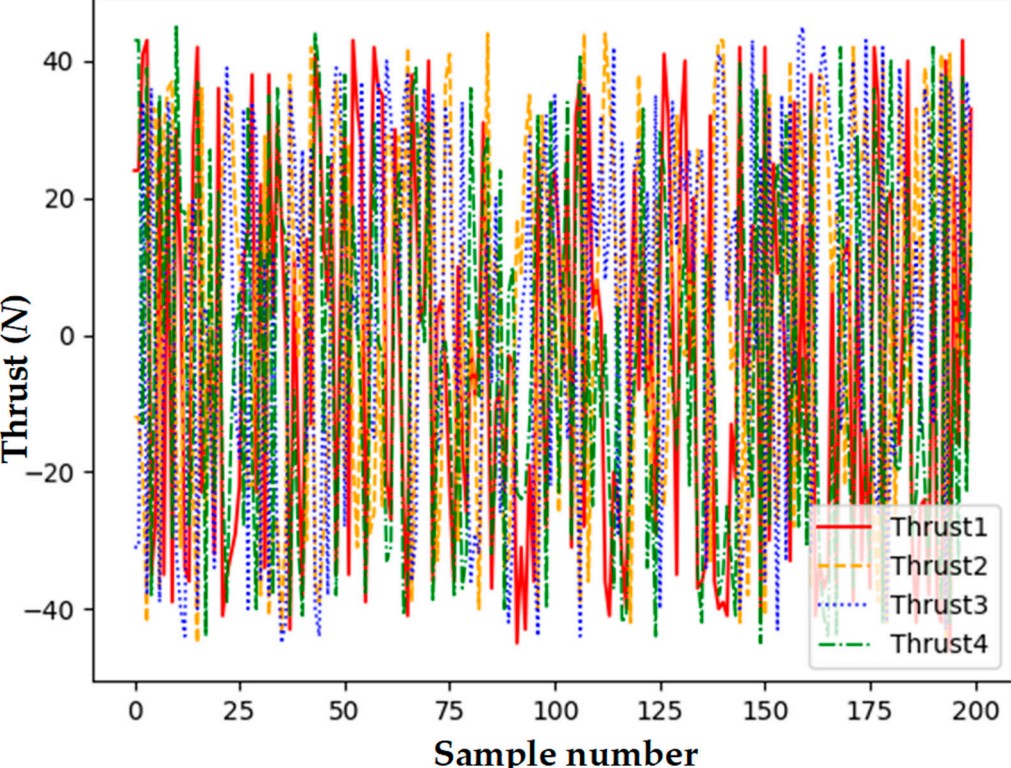

**Figure 4.** Heading priority thrust distribution results.

*2.2. Tunnel Autonomous Navigation*

In the geodetic coordinate system, the UUV starts from the initial position of the tunnel. To pass through the entire tunnel more safely and achieve the effect of detecting the inner wall of the tunnel, we set the desired heading as the axis direction of the tunnel pipe. The main research content of this article is that during the forward movement of UUV, when the heading changes due to other factors such as water impact, the vehicle can independently plan the heading based on the heading angle of the current position.

The deflection direction and deviation angle of the bow direction of the UUV in the tunnel relative to the central axis of the tunnel are essential parameters for how to plan the heading of the UUV at the next moment. This paper uses the combination conversion method of multiple-ranging sonar to obtain the above parameters. The calculation formula for ultrasonic ranging is as follows:

$$L = \frac{1}{2} \times v \times t \tag{13}$$

where $L$ represents the distance from the ultrasonic probe to the tunnel wall; $v$ represents the sound velocity value after temperature compensation; $t$ is the operating time of the sound wave within the measurement range.

Given a target point of UUV, by measuring and comparing the distances from both sides of the bow and stern of the UUV to the two walls of the tunnel, we can obtain the posture state of the UUV relative to the target point in the tunnel. Finally, we can adjust the control law of the heading PID controller to change the heading state of the UUV and always move towards the heading of the target point on the central axis. In this way, regardless of whether the vehicle's heading changes due to interference during navigation or the tunnel's forward direction changes, UUV can independently plan its heading based on its current heading angle, navigate toward the target point on the central axis, and achieve autonomous heading planning and control.

Figure 5 illustrates the autonomous heading planning process of UUV in the underwater tunnel detection task that is studied in this paper.

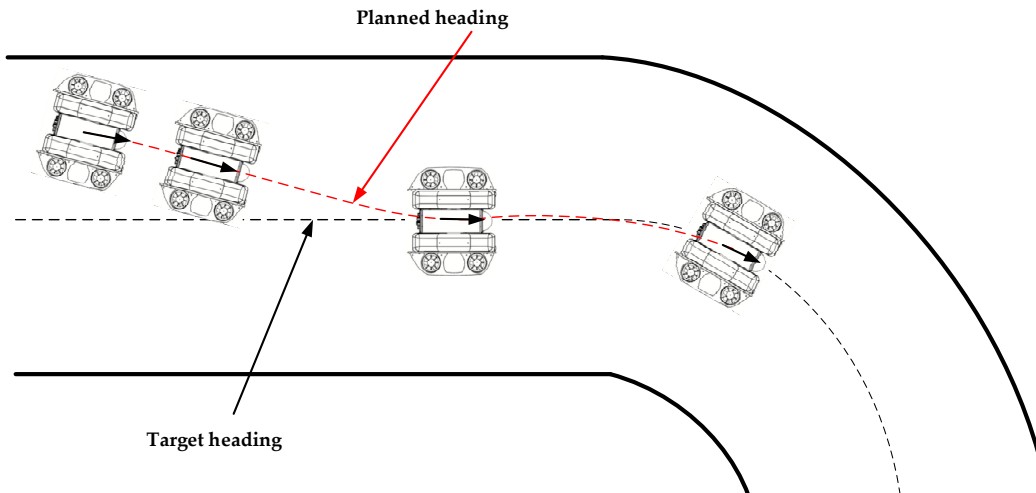

**Figure 5.** Schematic diagram of autonomous heading planning process.

The red curve in the figure shows the actual navigation trajectory of the UUV. Assuming that the exact position and target point at time t is $X(t) = [x(t), y(t), \varphi(t)]^T$, the autonomous heading planning problem of the UUV studied in this paper is to solve an optimal yaw control torque $N$ through the controller so that the error between the current heading angle $]\varphi(t)$ and the target heading angle $\varphi_T(t)$ converges to 0.

$$e(t) = \varphi_T(t) - \varphi(t) \tag{14}$$

When t tends to infinity, the error between the two heading angles approaches zero, we have:

$$\lim_{t \to \infty} e(t) = 0 \tag{15}$$

### 2.3. UUV Test Platform

To achieve autonomous heading planning and control of UUV, a new system is designed in this paper. The control system is based on the open-source underwater vehicle platform BlueROV, which is refitted and redesigned. NVIDIA and STM2 control board are combined to enable the vehicle's motion control and data acquisition. To provide heading feedback, a multi-ranging sonar sensor combination and an electronic compass are added. The sonar ranging sensor consists of four ultrasonic emission probes, installed on both sides of the vehicle's bow and stern. Finally, the UUV's control system is rebuilt based on the ROS system. Figure 6 shows the schematic diagram of the improved UUV platform hardware framework.

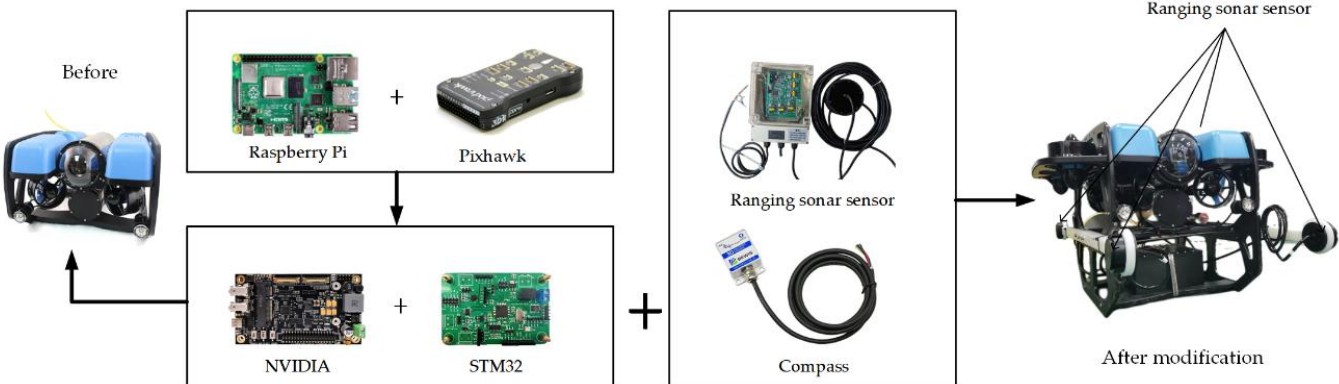

**Figure 6.** Schematic diagram of improved BlueROV hardware framework.

The modified UUV platform effectively addresses the issues of a bloated and redundant original control system architecture, and inconvenient redevelopment based on Pixhawk. Additionally, it attempts to integrate sonar ranging technology with the heading planning method to plan the UUV's heading through data feedback from the rangefinder. With this UUV platform, the designed autonomous heading planning and control system can be preliminarily verified.

The autonomous heading planning and control system is based on the ROS system. The functional modules of the UUV in the heading planning process are divided into different control nodes, including the upper computer interaction node; the lower computer heading planning node; the heading PID control node; the thrust distribution node; and the sensor data feedback node from top to bottom. Each functional node facilitates sensor data feedback and message transmission of motion control through topic communication [24]. Figure 7 shows the schematic diagram of the control system framework built in this paper.

In the control system designed in this paper, the autonomous heading planning and control process involves four main stages: target heading generation, heading planning, heading control, and heading feedback.

- The upper computer sends the target heading parameters to the lower computer through the interactive node. The heading planning node obtains the UUV's relative position in the environment and plans a desired yaw angle based on the feedback data from the ranging sonar sensor.
- The heading control node utilizes a PID controller to adjust the heading angle by distributing the thrust of the required rotation torque based on the planned yaw angle.
- The thrust distribution node thus obtains the thrust value and direction of each horizontal thruster and controls the heading of UUV to change. This process continues until the vehicle's forward heading is stabilized on the target heading.

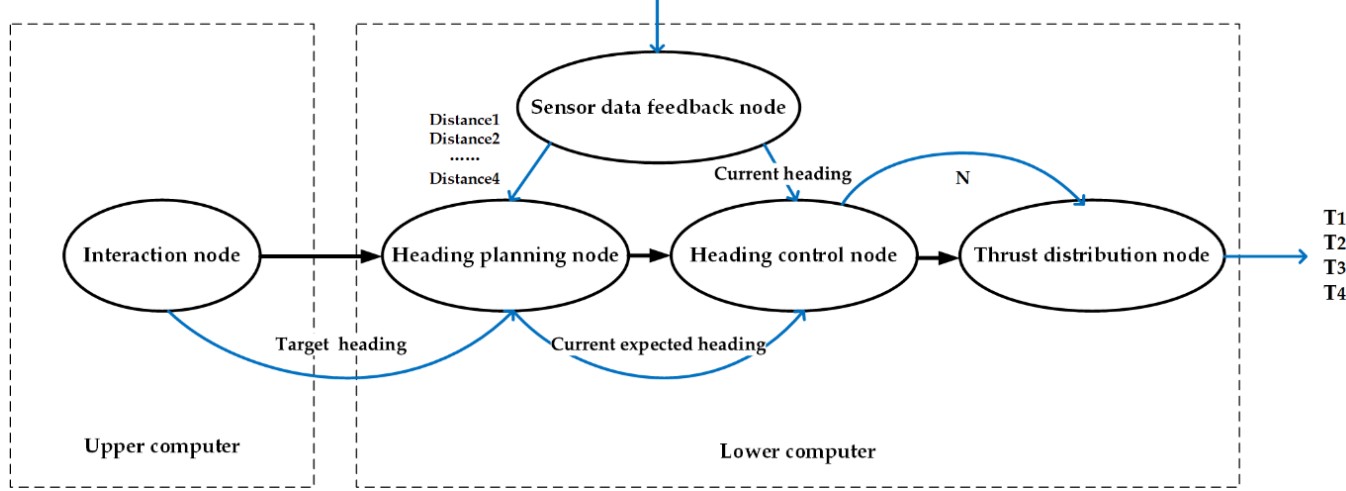

**Figure 7.** Schematic diagram of control system framework.

The flow chart of the control process is shown in Figure 8, where, $\varphi_d$ represents the current desired heading angle, $\varphi_T$ represents the target heading angle, and $\varphi$ represents the current heading angle.

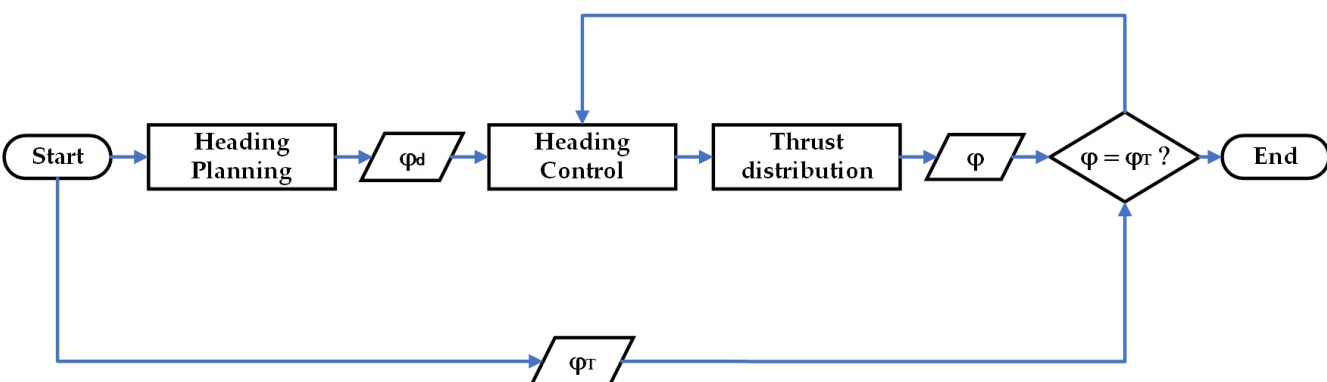

**Figure 8.** Flow chart of heading planning control.

The pseudocode flow of the proposed autonomous heading planning control Algorithm 1 is as follows:

---

**Algorithm 1** Heading Planning and Control

---

**1:** Set the initial parameters $P_T$ [1] and $R_r$ [2] of the heading planning method;
**2:** Initialize target heading angle $\varphi_T$;
**3: While** the procedure is in progress:
**4:**　　Calculate the yaw angle $\alpha_3$;
**5:**　　Send the desired heading angle $\varphi_d = \varphi + \alpha_3$ to heading PID controller;
**6:**　　**If** Current heading angle $\varphi \; != \varphi_T$:
**7:**　　　　Compute the control signal by heading PID controller;
**8:**　　　　Thrust distribution;
**9:**　　　　Send the control signal to the UUV;
**10:**　　　　$t++$;
**11:**　　**End if**
**12: End while**

---

[1] Distance from target point to initial position. [2] Radius of the tunnel.

## 3. Autonomous Heading Planning and Control Method

In this section, an autonomous heading planning method based on ranging sonar feedback control is designed, building upon the previously established hardware framework and control system. The ranging sonar uses ultrasonic spontaneous self-recovery to obtain the distance from the UUV's bow and stern to the two walls of the tunnel, allowing for the calculation of the deviation angle between the current and target headings [25]. By adjusting the UUV's heading through the heading control, the vehicle can achieve autonomous heading planning and control.

### 3.1. Heading Planning Model

Figure 9a below shows a simulation diagram of UUV navigating through a tunnel using the autonomous heading planning method proposed in this paper. As the tunnel is long and the direction is not constant, a planned distance is set and a target point is given to the vehicle on the target path. The distance from this point to the UUV's current position in the forward direction is referred to as the planned distance.

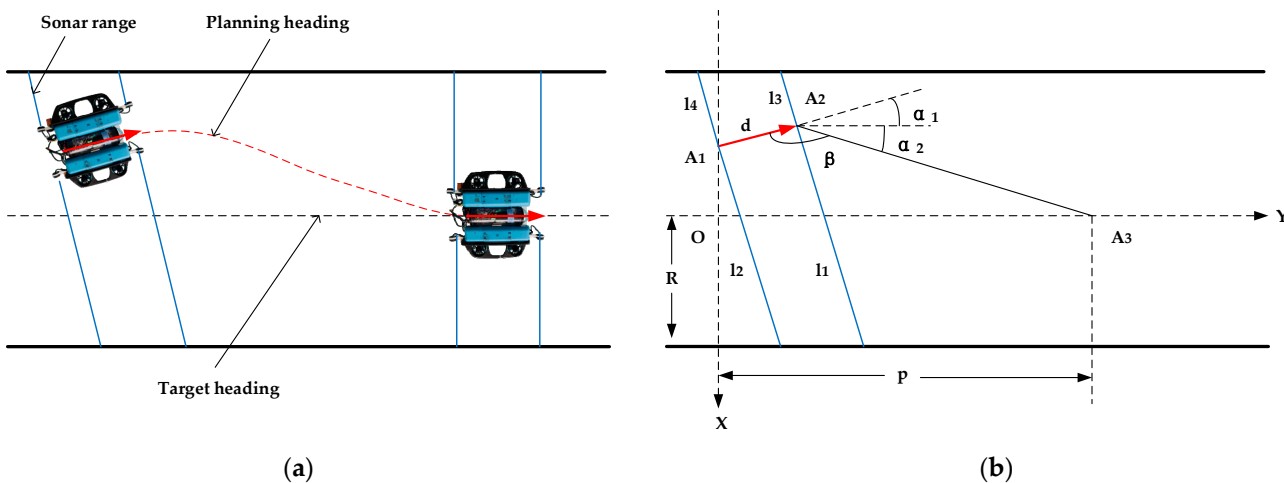

**Figure 9.** Model diagram of heading planning: (**a**) simulation process of autonomous navigation of UUV in tunnel; (**b**) mathematical model of autonomous navigation coordinate system.

To establish the model of heading planning, the central axis of the vehicle's center of gravity is taken as the standard and abstracted as a line. On this basis, establish a reference coordinate system based on the current position of the UUV, and the mathematical model is established as shown in Figure 9b.

In the established coordinate system shown in Figure 7b, let the origin be denoted as O. The length of the UUV body is denoted as d, the distances from the four ultrasonic probes to the pool wall measured by the ranging sonar sensor are denoted as $l_1, l_2, l_3, l_4$, and the distance from the tunnel axis to the two walls is denoted as R. The positions of the vehicle's bow and stern are represented by $A_1$ and $A_2$, respectively. $A_3$ represents the target point on the planned heading, and the planned distance from the target point to the origin of the coordinate system is denoted as $p$.

### 3.2. Autonomous Heading Planning Method

With the established mathematical model and coordinate system, we can express the coordinate forms of $A_1$, $A_2$ and $A_3$ through geometric transformation relations as follows:

$$A_1 : (-(l_2 \cos \alpha_1 - R), 0)$$
$$A_2 : (-(l_2 \cos \alpha_1 - R + d \sin \alpha_1), d \cos \alpha_1)$$
$$A_3 : (0, p)$$

With the coordinates of each point, the vehicle's current heading and planned heading can be expressed as vectors, representing the included angle between them. The deviation angle of the UUV's current heading relative to the target heading is denoted as $\alpha_1$ the deviation angle of the planned heading relative to the target heading is denoted as $\alpha_2$, and the included angle between the two vectors of the target heading and planned heading is denoted as β.

The deviation angle $\alpha_1$ can be solved using the sonar ranging sensor data on the same side and the length d of the UUV, based on the geometric relationship in the coordinate system:

$$\alpha_1 = \arctan\frac{l_1 - l_2}{d} \ or \ \alpha_1 = \arctan\frac{l_4 - l_3}{d} \tag{16}$$

From the solution formula of the angle between vectors, we can get:

$$\alpha_2 = \arccos\frac{\overrightarrow{A_2A_3} \cdot \overrightarrow{OA_3}}{\left|\overrightarrow{A_2A_3}\right| \times \left|\overrightarrow{OA_3}\right|} \tag{17}$$

$$\beta = \arccos\frac{\overrightarrow{A_2A_1} \cdot \overrightarrow{A_2A_3}}{\left|\overrightarrow{A_2A_1}\right| \times \left|\overrightarrow{A_2A_3}\right|} \tag{18}$$

where,

$$\overrightarrow{A_2A_1} = (d\sin\alpha_1, -d\cos\alpha_1)$$
$$\overrightarrow{A_2A_3} = (l_2\cos\alpha_1 - R + d\sin\alpha_1, p - d\cos\alpha_1)$$
$$\overrightarrow{OA_3} = (0, p)$$

Bring the above vectors into Equations (19) and (20) respectively to get:

$$\alpha_2 = \arccos\frac{p - d\cos\alpha_1}{\sqrt{(l_2\cos\alpha_1 - R + d\sin\alpha_1)^2 + (p - d\cos\alpha_1)^2}} \tag{19}$$

$$\beta = \arccos\frac{l_2\sin\alpha_1\cos\alpha_1 - R\sin\alpha_1 - p\cos\alpha_1 + d}{\sqrt{(l_2\cos\alpha_1 - R + d\sin\alpha_1)^2 + (p - d\cos\alpha_1)^2}} \tag{20}$$

From Figure 9b, it can be seen that the current heading of the UUV can be represented by the vector $\overrightarrow{A_1A_2}$. In order to quickly align the UUV with the target heading within the planned distance, the bow must be turned to the right by a certain angle so that the UUV heading aligns with the direction of vector $\overrightarrow{A_2A_3}$. This deflection angle is referred to as the desired deflection angle, denoted by $\alpha_3$, which is given by $\alpha_3 = \alpha_1 + \alpha_2$. To express the heading angle of the UUV at the current time in the established coordinate system, we denote it as $\varphi$. Then, the current desired heading angle can be obtained as $\varphi_d = \varphi + \alpha_1 + \alpha_2$.

Continuing from the previous paragraph, it is important to note that the above-discussed case is just one example of heading planning. After deducing and analyzing various possible heading states of the UUV in the tunnel, it has been concluded that the deflection direction of the current bow of the UUV with respect to the target heading can be determined based on the relationship between the magnitude of β and the complementary angle of $\alpha_1$. The deflection direction of the current bow then leads to the direction of $\alpha_3$ deflection, and the magnitude of $\alpha_3$ can be expressed as the sum of $\alpha_1$ and $\alpha_2$, as shown in Equation (21).

$$\alpha_3 = \begin{cases} |\alpha_1| + \alpha_2, & (\beta \leq 180° - |\alpha_1|) \\ |\alpha_1| - \alpha_2, & (\beta > 180° - |\alpha_1|) \end{cases} \tag{21}$$

The deflection direction of the UUV's bow relative to the target heading can be determined based on the magnitude of the included angle β and the complementary angle of $\alpha_3$. In order to simplify the calculation, we can artificially set the deflection direction of $\alpha_3 > 0$ as positive for right deflection and negative for left deflection. The deflection direction is not only related to the specific position of the UUV in the tunnel, but also to the direction of the current heading relative to the target heading. To judge the deflection direction of the UUV's bow, we can use the feedback data from the ipsilateral ranging sonar. If the distance from the bow to the wall on the ipsilateral side is larger than the distance from the stern to the wall, the UUV is left-deflected relative to the target heading, and vice versa. By deducing and reasoning about possible multiple heading states, we summarized and concluded the direction of $\alpha_3$, as shown in Table 3.

**Table 3.** Table of direction decision of $\alpha_3$.

| Case | Offset Direction of UUV Heading [1] | Direction of [2] |
|:---:|:---:|:---:|
| $l_1 > l_2$ | Left | $\alpha_3 > 0$ |
| $l_1 = l_2$ | Right, $\begin{cases} \alpha_2 > \alpha_1 \\ \alpha_2 < \alpha_1 \end{cases}$ | $\alpha_3 > 0$ <br> $\alpha_3 < 0$ |
| $l_1 < l_2$ | Level, $\begin{cases} l_1 > l_3 \\ l_1 < l_3 \end{cases}$ | $\alpha_3 > 0$ <br> $\alpha_3 < 0$ |

[1] The heading of UUV deviates from the direction of the central axis; [2] direction of heading planning angle.

After obtaining the magnitude and direction of $\alpha_3$, the current desired heading angle $\varphi_d$ is calculated by deflecting an angle of $\alpha_3$ in the specified direction from the current heading angle $\varphi$. That is, $\varphi_d = \varphi + \alpha_3$. This desired heading angle is then used as an input to the heading controller, which aims to control the UUV to deflect to the desired heading. As the UUV sails forward, the feedback data from the ranging sonar continuously changes, and the UUV plans and controls the heading by the current position and heading state until the vehicle finally stabilizes on the target heading.

During the UUV autonomous heading planning process, in addition to the accurate planning of the heading, there are also strict requirements for the control performance of the heading. In this paper, the algorithm expression of the PID controller for the heading control is given as follows:

$$N(t) = k_p \cdot e(t) + k_i \cdot \int_0^t e(t)dt + k_d \cdot \frac{de(t)}{dt} \tag{22}$$

where, $N(t)$ is the controller output signal that represents the planned rotational moment, $e(t)$ is the deviation signal of the controlled system, representing the deviation angle between the current heading of the UUV and the desired heading. The controller parameters are $k_p, k_i, k_d$.

## 4. Test and Analysis

The test platform in this section is the BlueROV rebuilt in Section 2.3. The main control board is an NVIDIA Jetson TX2 NX core module with a computing power of 1.33 TFLOPS. It is equipped with a carrier board from Realtimes Corporation, with a model of RTSO-6002E. The drive board is a self-made STM32 single-chip computer. The communication mode between the lower computers is serial communication. The software system is equipped with Ubuntu 18.04 operating system and the corresponding ROS Melodic system, using Python programming language, and the heading control algorithm is a classic PID control algorithm.

### 4.1. Heading Orientation Test Experiment

The heading planning control is a fundamental module in executing UUV autonomous navigation; it directly influences the effectiveness of the autonomous navigation control.

As such, it is crucial to verify the control performance of the UUV's heading control before conducting the autonomous heading planning test experiment. Figure 10 shows the procedure of the heading bow test experiment to obtain the control parameters $k_p, k_i, k_d$ of the heading PID controller. The desired heading referenced in the figure is the direction of the geographical coordinates 180°.

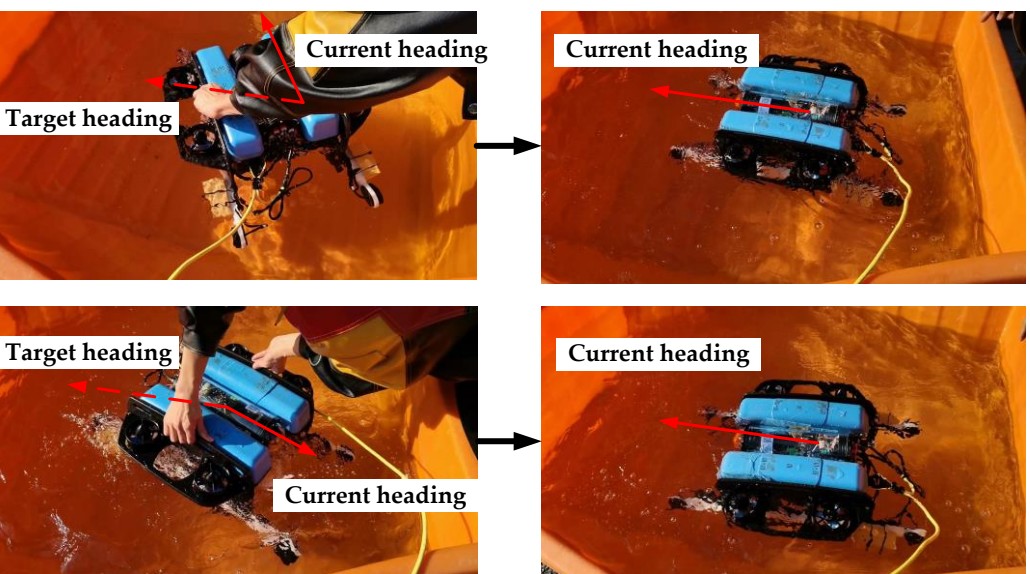

**Figure 10.** Process of heading orientation test experiment.

At the beginning of the experiment, the controller is given an initial value of $\delta$ roughly, and the UUV is artificially deflected to a certain angle with the target heading. After starting the heading bow function, the UUV automatically deflects to the target heading. The heading angle in the above process is stored. The following steps are adjusted to the PID control parameters by observing the critical oscillation curve using the critical proportionality method:

1.  First, use proportional control, starting from a larger proportionality $\delta$, and gradually reduce the proportional degree so that the system response to the step input can reach the critical oscillation state. The proportionality at this point is denoted as $\delta_r$, and the critical oscillation period is denoted as $T_r$;
2.  Determine the PID controller parameters according to the empirical formula of the critical proportionality method provided by Ziegler-Nichols (see Table 4); this method applies to the controlled object with self-balancing capability.

**Table 4.** Critical Proportionality Method for Setting PID Parameters.

| Controller Type | Proportionality $\delta\%$ | Integral Time $T_I$ | Differential Time $T_D$ |
|:---:|:---:|:---:|:---:|
| P | $2\delta_r$ | | |
| PI | $2.2\delta_r$ | $0.85T_r$ | |
| PID | $1.7\delta_r$ | $0.5T_r$ | $0.13T_r$ |

Figure 11 shows the heading angle change curve of the PID controller finally tuned through the critical proportionality method during the heading orientation experiment.

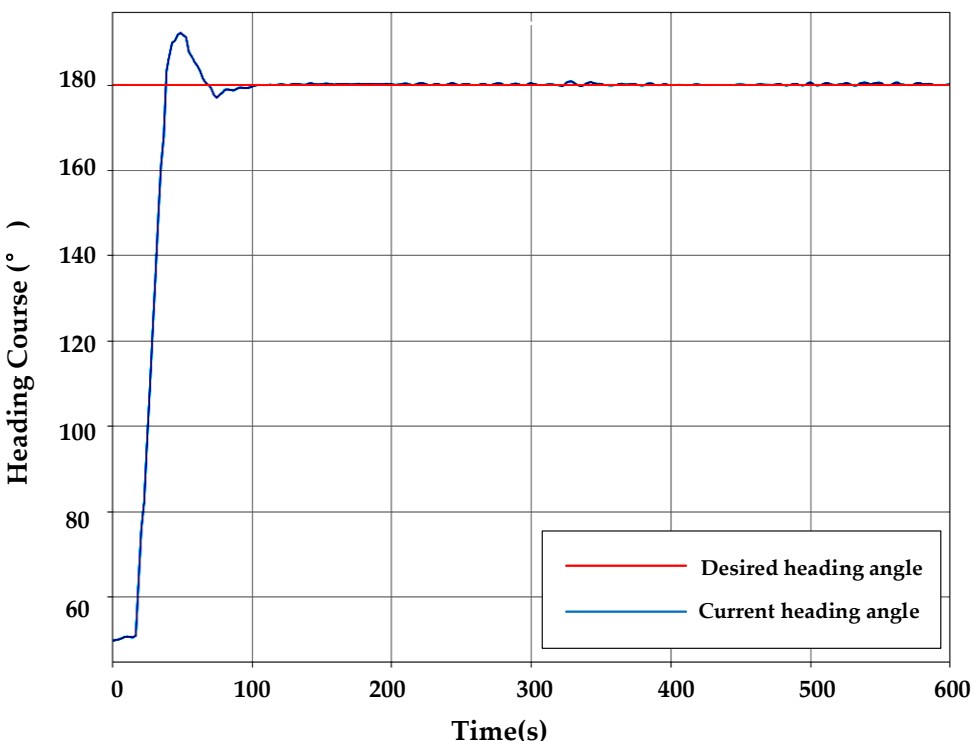

**Figure 11.** Test Results of PID Control for Heading orientation experiment.

As shown in Figure 11, the UUV quickly approaches near the target heading from the current heading angle, and promptly retraces to 180° in the opposite direction after the heading overshoot occurs, and finally remains stable at the heading angle of 180° to realize the UUV's heading orientation control. In this paper, the initial value of $\delta$ is set to 1. The following Table 5 shows the final obtained PID control parameters.

**Table 5.** Control parameters of the heading PID controller.

| Parameters | $k_p$ | $k_i$ | $k_d$ |
|---|---|---|---|
| Value | 0.530 | 0.400 | 0.175 |

*4.2. Autonomous Heading Planning Simulation and Verification*

We preliminarily verify the effectiveness of the proposed autonomous heading planning method by conducting simulation experiments in a simulation environment built with specific parameters. The parameters set for the simulation environment are: tunnel radius $R_r = 3.25$ and planning distance $P_T = 10$; and the target heading is set to the forward direction of the environment's central axis. The simulation experiment uses the simulated sonar acquisition data as the feedback input. Given a fixed speed $v_y = 1\ m/s$ for the UUV to move in the Y direction, the sampling time $t = 0.1\ s$. Changing the starting position of UUV in the simulation environment to simulate several situations that may occur when a robot navigates in a tunnel has been carried out. The following three sets of simulation experiments have been conducted, and the corresponding situations are as follows:

- Case I: The starting position of UUV is located at about 1 m to the left of the tunnel central axis, and the heading is left relative to the target heading;
- Case II: The starting position of UUV is located about 1 m to the right of the tunnel central axis, and the heading is right relative to the target heading;
- Case III: The starting position of UUV is near the central axis of the tunnel.

By recording the coordinate position changes of UUV in the simulation environment through autonomous heading planning during the simulation process, three sets of UUV path change curves for simulation experiments are drawn, as shown in Figure 12 below:

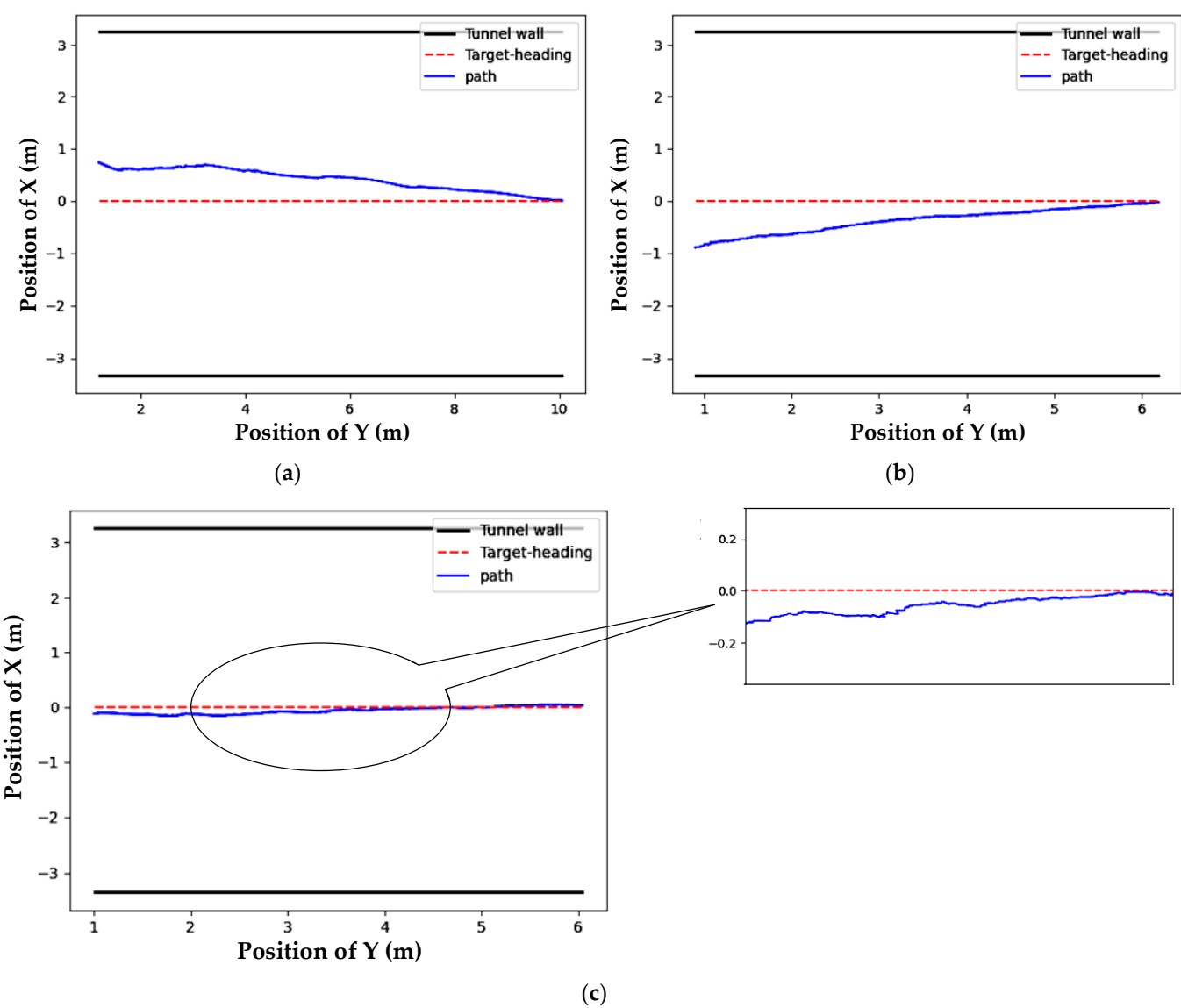

**Figure 12.** Path curve graph of UUV in simulation environment: (**a**) path curve of UUV in case I; (**b**) path curve of UUV in case II; and (**c**) path curve of UUV in case III.

From the above Figure, it can be seen that in the simulation environment, UUVs start from different starting positions. When the path where the target heading is located is far away, they can autonomously plan the heading to make the UUVs approach the central axis where the target heading is located and reach the target heading; when the UUV starts from or arrives near the path where the target heading is situated, the UUV can stabilize the heading within the error range of ±0.1 m for the target heading. It can be seen that the effectiveness of the proposed autonomous heading planning and control method for autonomous navigation control of UUV in a tunnel environment has been preliminarily verified.

*4.3. Autonomous Heading Planning Experiment and Analysis*

To better observe the effect of heading planning and control, we conducted test experiments in an indoor pool with a width of 6.5 m and a length of 160 m. The test environment was designed to simulate the navigation process of the UUV in a tunnel for tunnel detection, as shown in Figure 13.

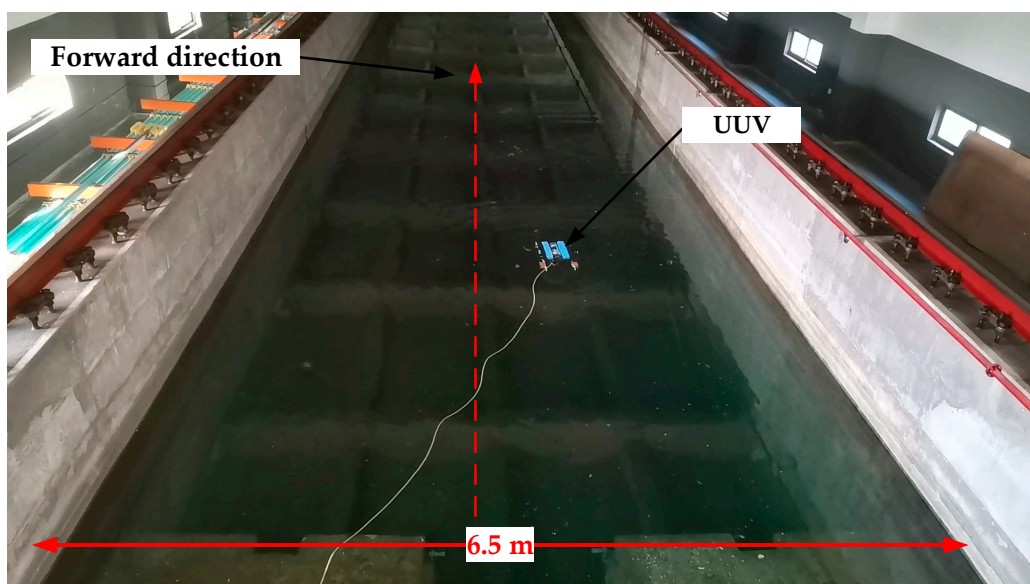

**Figure 13.** Test environment for autonomous heading planning experiment.

The UUV is initially placed at a position off the central axis, and its movement is controlled towards the forward direction from the starting point on the shore in the test environment shown in Figure 10. The scheme for autonomous heading planning is as follows: the UUV takes the forward direction of the central axis of the pool as the target heading. To observe the effect of heading planning, the UUV is manually placed at a position that deviates from the central axis. After starting the heading planning node, the heading change process of the UUV is carefully observed to determine whether it conforms to the expected effect. Figure 14 illustrates the heading change process in the test experiment.

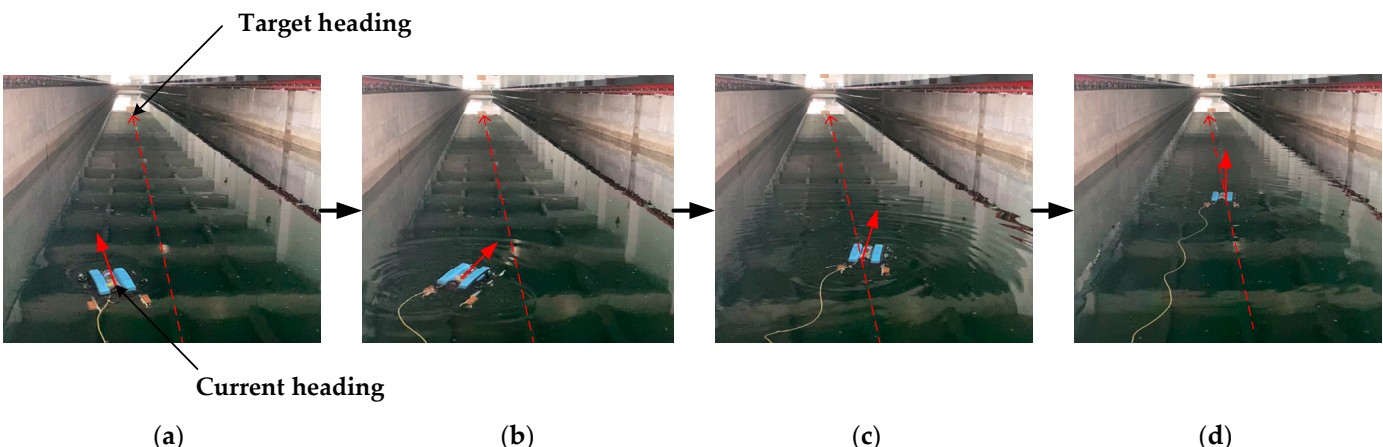

(**a**)          (**b**)          (**c**)          (**d**)

**Figure 14.** Process of UUV heading change in heading planning experiment: (**a**) UUV starts from the starting position; (**b**) UUV starts to deviate from the target heading; (**c**) UUV approaches the target heading; and (**d**) UUV steadily advances on the target heading.

The UUV in Figure 14a starts on the left side of the central axis of the pool, with its bow pointing to the left. After starting the heading planning node, the bow of the UUV in Figure 14b begins to deflect to the right, moving towards the central axis of the pool. As the UUV moves forward, the deviation angle between its current heading and the target heading in Figure 14c gradually decreases. Eventually, in Figure 14d, the UUV becomes stable near the central axis and advances steadily towards the target heading. It can be observed that the heading change process of the UUV satisfies the expected effect.

Based on the experiments conducted, it was determined that the UUV's target navigation path is along the central axis of the pool. Regardless of the UUV's starting position, the heading planning should guide the UUV towards the central axis. The ranging sonar is a critical sensor in the heading planning process, and its feedback can be used to analyze the control effect of heading planning. As the pool's width is 6.5 m, excluding the UUV's width, when the UUV reaches the target heading, the distance between the front and rear four sonar probes reaching the pool walls should be around 3 m. In a test conducted, we recorded the distance values from the four sonar probes to the pool walls as the UUV traveled forward, and their change process is shown in the figure below.

As depicted in Figure 15, the initial distances between the probes on the left and right sides of the UUV and the left and right walls of the pool are about 1.5 m and 4.5 m, respectively. As the UUV progresses forward, the distance between the probe on the right side and the right wall of the pool gradually decreases and eventually stabilizes at around 3 m. Conversely, the distance value of the probe on the left side gradually increases and finally stabilizes at approximately 3 m. This observation shows that the UUV, starting from a position on the left side of the central axis, deflected to the right until it finally stabilized on the target heading. Hence, the autonomous heading planning and control of the UUV produced satisfactory results.

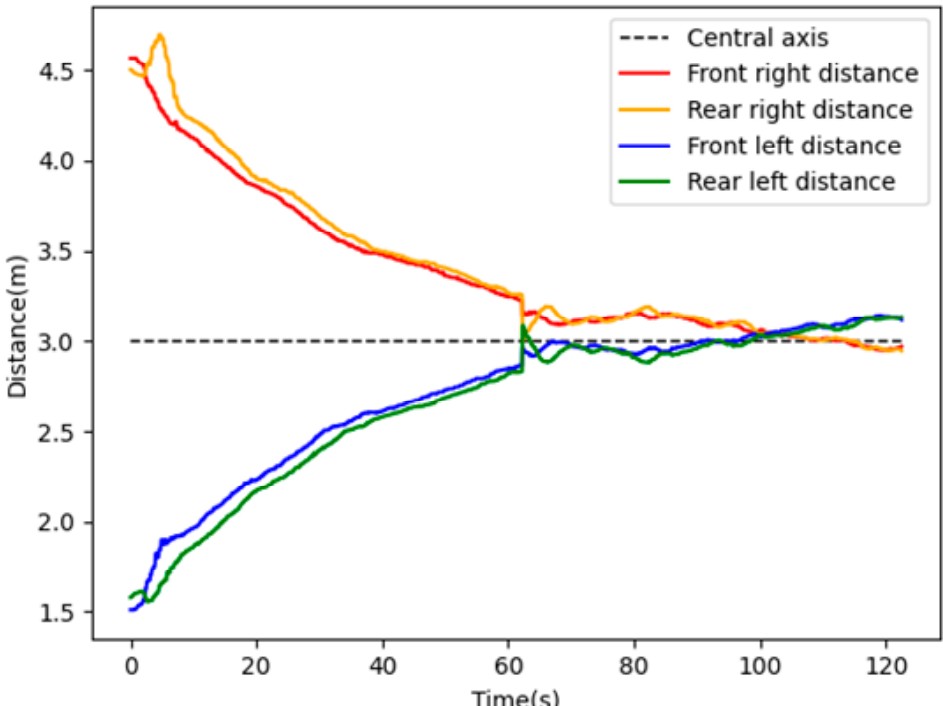

**Figure 15.** Range value change process of four probes of range sonar.

To further verify the effectiveness and stability of the proposed heading planning and control method, we conducted multiple sets of experiments to make the UUV start from different starting positions in the tunnel with different initial heading directions and recorded the change process of the current heading angle of the UUV during the experiment. Figure 16 shows the heading change curves for three of the experiments. The following figure shows the heading change curves in three sets of experiments, which correspond to the three cases in the above simulation experiments.

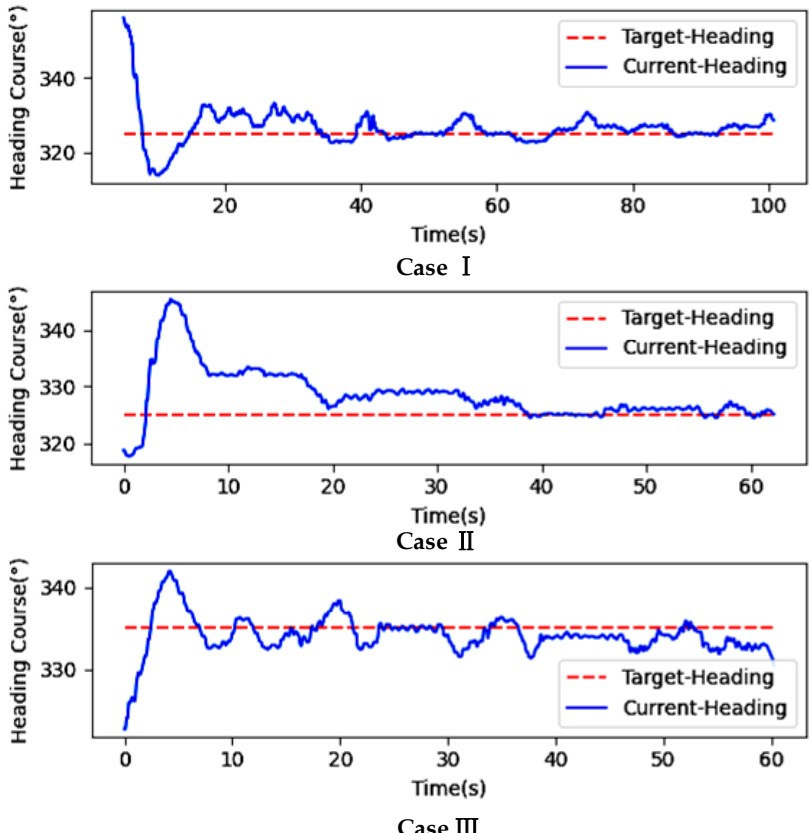

**Figure 16.** The change of heading angle during UUV navigation.

As can be seen from Figure 16, in Case I, the initial heading angle of UUV is about 350°, reaching the vicinity of the target heading at about 15 s and maintaining dynamic balance. In Case II, the UUV ultimately remains stable near the target heading of 325° as it navigates from a position where its prime angle is approximately 310°—and furthermore, it can be seen starting from different starting positions. Through autonomous heading planning, UUVs can approach the target heading faster and stabilize within a specific error range at the target heading angle. In Case III, the target heading is 335°, and the initial heading angle of the UUV is about 325°. Starting from a position close to the central axis of the tunnel, at an included angle with the target heading, it can be seen from the Case III that the UUV will dynamically adjust the current heading even after reaching the target path. It can be seen that when the heading is disturbed by water flow and affected by fluctuations, it can quickly adjust to the target heading again to achieve dynamic balance.

Figure 17 shows the variation of the desired deflection angle ($\alpha_3$) at each sampling time during the heading planning process in the three groups of experiments mentioned earlier.

Theoretically, when the UUV reaches the target heading, the heading should be kept as constant as possible, and the desired deflection angle should be equal to or tend to 0°. As can be seen from the Figure 17, in the three cases, the desired deflection angles for the start of UUV are 25°, 15°, and 10°, respectively, which is the deviation angle between the start heading of UUV and the target heading. Subsequently, the desired deflection angles are gradually reduced to 0° after the first arrival at the target heading. It can be seen that starting from different positions, $\alpha_3$ of UUV changes from large to small, approaching the target path with a large deflection angle and approaching 0° after reaching the vicinity of the target path. Finally, it adjusts up and down in a small range, consistent with our planning effect.

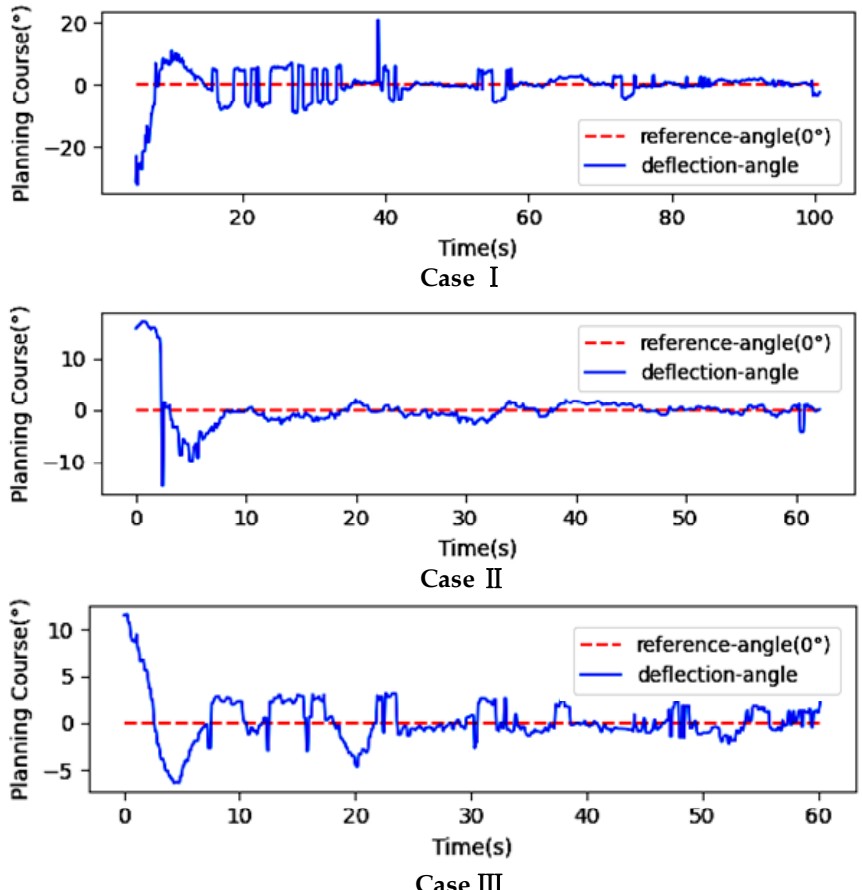

**Figure 17.** Changes of during UUV navigation.

After observing the experimental data, it can be seen that the UUV heading is stable near the target heading after the experiment lasts for about 20 s. To further analyze the control performance of the heading planning method near the target heading, the deviation between the current heading of the UUV and the target heading during the experiment is analyzed numerically; the results are shown in Table 6.

**Table 6.** Table of heading deviation related data.

| Test | Max Deviation [1] (°) | Mean Deviation [2] (°) | Standard Deviation [3] (°) |
|------|------------------------|------------------------|-----------------------------|
| Case I | 5.925 | 1.049 | 1.829 |
| Case II | 4.428 | 1.688 | 1.512 |
| Case III | 4.583 | −1.081 | 1.249 |

[1] Absolute value of the maximum heading deviation from the target heading; [2] mean deviation from target heading; [3] and standard deviation from target heading.

The table above shows the maximum absolute deviation of UUV from the target heading in each experiment, which were 5.925°, 4.428°, and 4.583°, respectively. The average deviation of the heading deviation angle during the navigation process was calculated and found to be stable within ±2° of the target heading. Using the target heading as the allowable error, the reference standard deviation of the heading deviation was set to 3. The standard deviations of the heading deviation angles in the 3 experiments were calculated to be 1.829°, 1.512°, and 1.249°, which were all much less than the reference standard deviation. This indicates that the dispersion of the UUV's heading deviation after approaching the target heading is small, and can be well-stabilized in a slight deviation range. Therefore, it can be concluded that the autonomous heading planning control

method proposed in this paper has good control performance, and strong applicability for the heading control process.

## 5. Conclusions

This paper investigates the problem of autonomous heading planning and control of UUV for underwater missions in a closed pipeline environment such as a tunnel. During the autonomous underwater mission, the UUV is required to maintain a safe passage through the tunnel and other environments on the path where the target heading is located and to be able to adjust and control the heading autonomously when there is a change in the disturbed heading or a change in the tunnel direction. This paper proposes an autonomous heading planning and control method based on sonar ranging feedback control, designs an original algorithm for UUV heading planning through the combination of multiple sonars ranging sensors, and combines it with a heading PID control method, which simplifies the structure and computational complexity of the control algorithm while weakening the dependence on the system model and external parameters. It is not necessary to obtain the path reference points in advance, but to plan and control the heading online according to the actual state of the UUV. Finally, simulation validation and pool tests show that the proposed method is responsive and adaptable in heading planning and control, and can effectively solve the problem of autonomous heading planning and control of UUVs in tunnels and other environments. In addition, although the method is inclusive of the obstructive influence of the motion, how to solve the problem of the effect of the more extensive water flow impact on the control stability—as well as the effect of the environment on the feedback of the sonar data, combining the method with robust control, fault diagnosis fault tolerance algorithms, etc.—to make the whole system more stable, is our next research direction.

**Author Contributions:** Conceptualization, D.C., Z.C., T.X. and X.Y.; methodology, D.C. and Z.C.; software, T.X. and X.Y.; investigation, T.X. and X.Y.; writing—original draft preparation, T.X.; writing—review and editing, D.C. and Z.C.; visualization, T.X.; supervision, D.C. and Z.C. All authors have read and agreed to the published version of the manuscript.

**Funding:** This research was supported in part by the National Natural Science Foundation of China (No. U2006228, No. 52171313, No. 51839004, No. 52101362), the High tech ship innovation project (CY04N20) and the Key Laboratory Foundation for Underwater Robot Technology (No. 6142215200305).

**Institutional Review Board Statement:** Not applicable.

**Informed Consent Statement:** Not applicable.

**Data Availability Statement:** Not applicable.

**Conflicts of Interest:** The authors declare no conflict of interest.

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
