# Peer review of "Autonomous Heading Planning and Control Method of Unmanned Underwater Vehicles for Tunnel Detection"

_jmse, doi:10.3390/jmse11040740_

Round 1

Reviewer 1 Report

This paper presents an autonomous heading planning and control of UUV for long-distance underwater tunnel detection. Experimental results via pool tests are provided to demonstrate the performance of the presented methods. Please carefully address the following comments:

1. What is the specific control method used in the design (eg., adaptive, robust, pid, fuzzy, etc)?. This is not well described in this paper, and not well-informed in both abstract and introduction.

2. The sonar ranging feedback is not the main methodology, since it is simply provide the state (measurement) to the controller. Please enhance more on the control method.

3. What does AUV refers to in line [55]?.

4. No comparison is provided in this paper. Comparison via simulation can be provided to highlight the difference the previous relevant work. 

5. The following reference would be useful to broaden the prespectives about heading/tracking control problem using variable structure based sliding mode control schemes:

Discrete-time modified repetitive sliding mode control for uncertain linear systems

6. In term of languages, this paper uses some uncommon words that somehow make the messages are hard to grasp. More proofreading is required on this paper.

Reviewer 2 Report

In my opinion, it is a study to deal with a necessary problem, however, there are a lot of things that need to be addressed to meet the quality publication. Some of the concerns are as follows:

1/ Some abbreviations, especially SMC, AUV, ROS and PID in the paper, should be spelled out when they are introduced.

2/ Despite the motivating topic, the novelty of the approach of the manuscript does not seem significant since this manuscript simply presented the heading control of ROV using a simple PID controller. This paper should have more theoretical contributions. Although it is related to the marine system control strategy, the selection of parameters is not discussed in detail, and the explanation of the procedure and contents of the manuscript are ambiguous.

3/ It will be better if the authors can also evaluate the performance of your algorithm through simulation results under specific environmental parameters, and compare it with the experimental results.

4/ The presentation of the paper can be improved and the quality of some figures can be enhanced. It is hard to see the quality of the figures in the manuscript. They are quite grainy but this may be a function of the PDF process. The quality of the figures needs to be improved such as: Figure 6, Figure 8b, Figures 12 and 13.

5/ In the introduction part, I think that the authors could enrich the reference section by discussing the concept of sliding mode control should be discussed, some new works related to sliding mode control of marine vehicles or second order systems, especially the dynamic sliding mode control methods, robust sliding mode method, multiple sliding mode methods and so on, should be included. To help the authors in this direction, I suggest the following reference, robust position control of an over-actuated underwater vehicle under model uncertainties and ocean current effects using dynamic sliding mode surface and optimal allocation control, station-keeping control of a hovering over-actuated autonomous underwater vehicle under ocean current effects and model uncertainties in horizontal plane, design of a non-singular adaptive integral-type finite time tracking control for nonlinear systems with external disturbances, perturbation observer-based robust control using a multiple sliding surfaces for nonlinear systems with influences of matched and unmatched uncertainties, finite-time convergence of perturbed nonlinear systems using adaptive barrier-function nonsingular sliding mode control with experimental validation, fast terminal sliding control of underactuated robotic systems based on disturbance observer with experimental validation, adaptive nonsingular terminal sliding mode control for performance improvement of perturbed nonlinear systems. And the introduction should be added to do a better job of explaining the existing methods and why they are or are not valuable. The authors reviewed and summarized some existing methods relating to their work. What research gap did you find from previous researchers in your field (it is still partially described, but needs to be expanded and made clearer)?  Mention it in the Novelties section. It will improve the strength of the article. The motivation and background of wide practical use of the theoretic results presented should be clearly emphasized to facilitate the readers.

6/ In section 2.1, a sub-section “Assumptions” should be added to make the problem clearer. All assumptions and physical constraints should be provided.

7/ Also, the kinematic and dynamic model of the ROV need to be added in section 2 of the manuscript. The author can refer to study on dynamic behavior of unmanned surface vehicle-linked unmanned underwater vehicle system for underwater exploration.

8/ The content of Section 2.2 “Tunnel autonomous navigation” is very limited and not clear. The theory of the method applied in the manuscript should be described carefully. The authors have to show the manner of implementation in professional details which will be beneficial to the readers.

9/ Why didn’t the author use the Line of sight (LOS) algorithm for heading planning? What is the difference between your algorithm and LOS? What are the novelties of the proposed method? What are the underlying factors that led to better performance of the proposed method? Relate to the LOS algorithm, please see this, robust position control of an over-actuated underwater vehicle under model uncertainties and ocean current effects using dynamic sliding mode surface and optimal allocation control.

10/ Please add more details of how the parameters of the controller k_p, k_i, k_d are obtained.

11/ The considered problem is simplified. no model uncertainties or disturbances are considered. Obviously, the performance of underwater vehicle is affected by environmental disturbances such as wind, wave, and current. There is no result robustness under the disturbance. The author needs to give more detailed data references or results.

12/ Detailed implementation information should be provided (hardware, software, configuration, settings).

13/ More important results of the ROV should be added. The reviewer does not see any experimental results of the ROV such as: the experimental results of control input Fx, Fy, N and 4 thrusters T1, T2, T3, T4 in Equation 3. Also, the explanations and analysis of simulation results should be enriched to show the validity of data.

14/ Figures 12 and 13 showed the results in 3 cases: I, II, and III. What are the conditions of the 3 cases? Please clarify it.

15/ Line 378, Figure 12 or Figure 13?

16/ In results part, more design parameters and comparisons with some existing results are recommended. Compared with existing results, the advantages of the manuscript should be further highlighted and the occurred difficulties of conducted topic can be explained. A discussion on robustness of the proposed algorithm with respect to disturbance is needed. A discussion on robustness of the proposed algorithm with respect to system or sensor failures is needed.

17/ Please rewrite the conclusion of this manuscript. The Conclusion section is superficial, should include quantitative results, advantages and disadvantages, limitation and recommendation for new implementations and future work. The novelty and contributions of the proposed system should be emphasized. The objective/problem statement need to be explained.

18/ The manuscript writing can be further polished with professional English. The manuscript can be thoroughly revised for grammar check. There are some unclear sentences along the paper.  Some typo errors, lines 87-93, “Part” should be “Section”, line 95, “mathematical model” should be “Mathematical model”, line 408, “We have…” should be “we have..” and so on. Please check the entire manuscript

Round 2

Reviewer 1 Report

This paper has been well revised and is acceptable for publication. Just minor comment: please check refs [13] and [14] in References section seems incorrect. 

Author Response

Thank you very much for your affirmation and suggestions regarding this revised manuscript. After carefully reviewing the references [13] and [14], we found that some of the information was incorrect due to our negligence, and the relevant information has been revised. Thank you again for your kind reminder.

Reviewer 2 Report

Thank you for the revised manuscript. I appreciate your efforts to revise the manuscript in light of the comments addressed in the previous review. Most of the recommendations I made were addressed by the authors. This paper version now is better than before. However, the format of references needs to be unified with the aim to satisfy the requirement of the journal, and the DOI number of some new references needs to be added to this paper.  The author’s name of some references is wrong and needs to be rearranged such as References 7, 11, 12, 13, 14, 22, and so on. Please check carefully.

Author Response

Thank you very much for your affirmation of this revised manuscript and for your reminder of the format of the references. After carefully reviewing the references [7], [11], [12], [13], [14], and [22], it was found that some of the author's names were incorrectly sequenced and spelled due to our negligence. The relevant information has been modified in the corresponding references. Thank you again for your kind reminder.